# Environmental Planning and Non-Communicable Diseases: A Systematic Review on the Role of the Metabolomic Profile

**DOI:** 10.3390/ijerph20146433

**Published:** 2023-07-23

**Authors:** Natália Cristina de Oliveira, Pedro Balikian Júnior, Arnaldo Tenório da Cunha Júnior, Edson de Souza Bento, Josealdo Tonholo, Thiago Aquino, Filipe Antonio de Barros Sousa, Gustavo Gomes de Araujo, Maurício Lamano Ferreira

**Affiliations:** 1Department of Nursing and Health, Guarulhos University, Central Campus, Guarulhos 07023-070, SP, Brazil; 2Institute of Physical Education and Sport (IEFE), Federal University of Alagoas, Campus AC Simões, Maceió 57072-900, AL, Brazil; balikianpedro@gmail.com (P.B.J.); filipe.sousa@iefe.ufal.br (F.A.d.B.S.); gustavo.araujo@iefe.ufal.br (G.G.d.A.); 3Kineanthropometry, Physical Activity and Health Promotion Laboratory (LACAPS), Physical Education Department, Federal University of Alagoas, Campus Arapiraca, Arapiraca 57309-005, AL, Brazil; arnaldo.junior@arapiraca.ufal.br; 4Institute of Chemistry and Biotechnology (IQB), Federal University of Alagoas, Campus AC Simões, Maceió 57072-900, AL, Brazil; edson.iqb@gmail.com (E.d.S.B.); tonholo@gmail.com (J.T.); thiago.aquino@iqb.ufal.br (T.A.); 5Department of Geoenvironmental Analysis, Guarulhos University, Central Campus, Guarulhos 07023-070, SP, Brazil; mauecologia@gmail.com

**Keywords:** metabolomics, environmental hazards, non-communicable diseases, environmental planning

## Abstract

Non-communicable diseases (NCDs) are the major cause of death worldwide and have economic, psychological, and social impacts. Air pollution is the second, contributing to NCDs-related deaths. Metabolomics are a useful diagnostic and prognostic tool for NCDs, as they allow the identification of biomarkers linked to emerging pathologic processes. The aim of the present study was to review the scientific literature on the application of metabolomics profiling in NCDs and to discuss environmental planning actions to assist healthcare systems and public managers based on early metabolic diagnosis. The search was conducted following PRISMA guidelines using Web of Science, Scopus, and PubMed databases with the following MeSH terms: “metabolomics” AND “noncommunicable diseases” AND “air pollution”. Twenty-nine studies were eligible. Eleven involved NCDs prevention, eight addressed diabetes mellitus, insulin resistance, systemic arterial hypertension, or metabolic syndrome. Six studies focused on obesity, two evaluated nonalcoholic fatty liver disease, two studied cancer, and none addressed chronic respiratory diseases. The studies provided insights into the biological pathways associated with NCDs. Understanding the cost of delivering care where there will be a critical increase in NCDs prevalence is crucial to achieving universal health coverage and improving population health by allocating environmental planning and treatment resources.

## 1. Introduction

Metabolomics is a rapidly evolving field that deals with the assessment of metabolite changes in response to endogenous and/or exogenous perturbations [1] as well as assists in the identification of biomarkers linked to emerging pathologic processes [2]. Metabolites are important intermediates and end products of metabolism, indicators of processes underlying a given disease, and can predict the responses of such conditions to therapeutic interventions [3]. In this sense, metabolomics is a useful tool, but it is still little applied in the diagnostic and prognostic processes relating to NCDs [3].

The increasing burden of NCDs is a relevant threat for the population worldwide. In addition to being the most major cause of death, they have economic, psychological, and social impacts [4]. The indirect costs of these diseases (loss of productivity, cost of caregivers, etc.) in developed countries exceed by five times their direct costs (treatment and hospitalization) [4]. In resource-limited settings, as in the case of low- and middle-income countries, treatment of NCDs is even more challenging due to their overlap with the burden of infectious diseases [5].r

Numerous cases of NCDs have historically been associated with environmental variables. Among them, air pollution in medium and large urban centers has played a leading role in triggering inflammatory processes. Air pollution is the second most major cause contributing to NCDs-related deaths globally [6]. Human exposure to fine particles (PM2.5) is one of the main health concerns linked to mortality [7]. Reductions in air pollution, especially PM2.5, contribute to longevity gains and numerous other health benefits [8].

Developing countries experienced rapid urban development throughout the 20th century, and, consequently, severe challenges are faced by current populations. Among them, serious public health problems have arisen from a lack of strategic urban planning and health services that are essentially focused on the treatment of chronic diseases instead of promotion of early diagnosis, which would enable environmental planning for the territory to build a health promotion scenario. Thus, intersectoral and cross-disciplinary research combining health and environmental planning may contribute to increasing awareness and implementing solutions to this problem [9].

Efforts to improve the prevention and management of NCDs are part of the current international agenda [10]. Target 3.4 of the Sustainable Development Goals (SDGs, from the United Nations 2030 Agenda) aims to reduce premature mortality from NCDs by one third through prevention and treatment. A shared feature of all NCDs is chronic low-grade inflammation, promoted by unhealthy diets, environmental pollutants, microbial exposure, and psychological and biological stress [11]. Hence, acknowledging tools that provide early diagnosis of the biomarkers associated with this condition is imperative to effectively reaching this target.

Despite the publication of a mini review focusing on five conditions in which metabolomic tools have been employed for diagnosis, treatment, or prognosis prediction [3], no comprehensive systematic reviews were found covering this topic in the Cochrane Library database (May 2023). Furthermore, the redesign of urban areas to promote an active lifestyle and a healthier environment [12] and territory ordering that considers public health issues are key elements to promote equity in the achievement of SDGs.

Thus, the aim of the present study was to review the scientific literature on the application of metabolomics in NCDs and to discuss environmental planning actions to assist healthcare systems and public managers based on early diagnosis.

## 2. Materials and Methods

The present study was conducted following the “Preferred Reporting Items for Systematic Reviews and Meta-Analyses” guidelines (PRISMA—https://prisma-statement.org/, accessed on 15 June 2023). The criteria employed in the search and presentation of results were based on the PICO approach: population, intervention, comparison, and observation or result [13].

The study protocol awaits PROSPERO registry number (receipt 446279), and Prisma checklist is available (see Appendix A).

Databases consulted were “Web of Science”, “Scopus”, and “PubMed”. No publication date limit was set, and only studies in English involving humans of all ages were included. The search strategy involved the following MeSH terms: “metabolomics” AND “noncommunicable diseases” AND “air pollution”; “metabolomics” AND “noncommunicable diseases”.

Review studies, protocols, comments, positions, or guidelines were excluded from this study. After the search, each publication was evaluated for information on the outcome of interest (metabolomic profile in the prediction of non-communicable diseases).

Furthermore, a discussion on environmental guidance for territory planning considering estimated public health issues was proposed.

## 3. Results and Discussion

The first search (“metabolomics” AND “noncommunicable diseases” AND “air pollution) resulted in zero publications after application of the exclusion criteria. We then proceeded to the second search (“metabolomics” AND “noncommunicable diseases”), which resulted in 80 publications (PubMed n = 8, Scopus n = 25, and Web of Science n = 47). After checking for duplicate records and applying the exclusion criteria, 12 studies were dismissed, and 39 were excluded for other reasons after abstract or full text reading (Figure 1). Thereby, a total of 29 studies were eligible for the present review (PubMed n = 5, Scopus n = 7, Web of Science n = 17).

Despite the absence of a publication date limit in the search, the oldest paper included in the present review was published in 2013, highlighting that this issue represents a quite novel field of scientific investigation. Most studies were conducted in Iran (n = 7) and in the UK (n = 6), but South Africa (n = 4) and China (n = 3) also had a number of studies covering this subject, as did Finland and Australia (two studies each) and Denmark, the Netherlands, Slovenia, Estonia, Spain, the USA, and Ghana, with one publication each (Figure 2).

Eleven studies involved NCDs prevention (Table 1), and, among them, five studied children or adolescents [14,15,16,17,18]. The remaining investigations focused on the comparison of lifelong athletes and sedentary subjects [19], trained and untrained individuals [20], night shift and day shift workers [21], a dietary intervention aimed at preventing NCDs [22], a retrospective study with patients who were hospitalized or died from NCDs [23], and a follow-up study on usual alcohol consumption [24].

Breastfeeding duration seems not to influence metabolomic profile after 20 years; rather, current lifestyle and environmental factors have a stronger potential influence [14]. Nonetheless, there is evidence of an association between parent and child metabolite concentration in plasma [15]. Genetics, eating habits, and physical activity may explain the association of parents’ body mass index (BMI) with the higher BMI and waist circumference of their offspring at 20 years [14].

Overeating behavior in childhood seems to produce a metabolic profile characterized by chronic inflammation and higher lipid concentrations, a risk factor for NCDs [17]. When comparing children with a normal weight with their overweight or obese counterparts, some fecal metabolites allow differentiation between groups [18]. This may indicate that the intestinal microbiota and its metabolites play an important role in the development of obesity.

Alterations in a common component of triglyceride-rich lipoproteins, APOC3, a small 99-amino-acid peptide, are associated with a substantial decrease in the risk of coronary artery disease, reduction in triglyceride, and changes in VLDL and HDL levels. A follow-up study with more than 13,000 subjects, including children, mothers, and elderly women, was able to confirm and characterize these associations, useful for assessing drug targets in dyslipidemia [16].

Changes in metabolic profile were also verified among night shift workers [21]. They support the association between night shift working and many common NCDs. Opposite to this finding, Orrú et al. [19] evaluated muscle biopsies of lifelong athletes and age-matched untrained subjects. The authors found that physical activity significantly influences the expression of proteins and metabolites involved in the reduction of the onset of NCDs. In fact, only 10 days of aerobic exercise already allows distinction between trained and untrained individuals, assessed by urine metabolomics [20], reinforcing the role of regular exercise in the prevention of NCDs.

Diet also seems to influence urinary metabolomics. Biomarkers from healthy foods are significantly higher after the consumption of a diet following WHO guidelines for prevention of NCDs [22]. Low-income countries suffer from problems of child malnutrition and lack of nutritional quality in school meals [25]. In this sense, the metabolomic profile can play an important role as an effective monitoring and diagnostic tool for understanding nutritional problems and proposing effective public policies to attend to this population.

A cohort study with patients who either died or were hospitalized with NCDs revealed that more than 400 metabolites are common to at least two NCDs [23]. Risk factors such as smoking and low-grade inflammation were identified as antecedents of NCD multimorbidity and have the potential for early prevention.

Another risk factor for NCDs is increased alcohol intake. This behavior was studied by Würtz et al. [24], who observed that it is associated with cardiometabolic risk markers across multiple metabolic pathways. Table 2 presents eight studies involving diabetes [26,27,28,29], insulin resistance [30], systemic arterial hypertension [31,32], and metabolic syndrome [33].

There seems to be a causal link between red meat intake and type 2 diabetes (T2D) as red meat metabolite score is associated with T2D incidence and potentially with other cardiometabolic diseases [26]. Another eating habit associated with T2D incidence is food neophobia (a behavioral characteristic in which a person withdraws from tasting unfamiliar or new foods). Food neophobia may also by associated with coronary heart disease [29], as this habit is related to health biomarkers.

Some amino acids and acylcarnitines are considered potential risk markers for diabetes as they reflect disturbances in several metabolic pathways among diabetic individuals [28]. Metabolic patterns such as higher levels of leucine and its catabolic intermediates are useful for identifying and monitoring T2D risk prior to disease onset [27].

Insulin resistance (IR) also presents a specific metabolic signature, identified by Arjmand et al. [30]. When compared to healthy individuals, persons with IR present an increase in branched-chain amino acids (valine and leucine), aromatic amino acids (tyrosine, tryptophan, and phenylalanine), alanine, and free carnitine.

As well as in diabetes, high plasma levels of carnitines and acylcarnitines seem to play a crucial role in the development and progression of systemic arterial hypertension, together with low plasma levels of glycine [32]. Hypertensive patients present a distinctive metabolomic profile when compared to normotensive subjects, mainly characterized by alterations in branched-chain amino acid (BCAA) metabolism. Strauss-Kruger et al. [31] conducted a study on masked hypertension (normotensive at the doctor’s office but hypertensive out of it) and observed that these changes in BCAA metabolism may be modulated by central adiposity, indicating that metabolic dysfunction may be an underlying contributor to the etiology of hypertension.

A group of interrelated cardiometabolic abnormalities, including hypertension, hyperglycemia, dyslipidemia, and central adiposity, and metabolic syndrome (MetS) have also been studied with a metabolomic approach. The alteration in circulating levels of amino acids and acylcarnitines is related to the increase in the number of MetS components [33].

Six studies focused on obesity [2,34,35,36,37,38,39] (Table 3). Intake of sugar-sweetened beverages was positively associated with obesity-related markers [36], and a set of 9 amino acids and 10 polar lipids may also be a potential biomarker of adult obesity [38]. Some metabolites are able to discriminate metabolically healthy obesity from metabolically unhealthily obese subjects [2]. Regarding this latter issue, a particular pattern of amino acids and choline-containing phospholipids may assist in the identification of metabolic health among obese patients [2].

Also noteworthy is the influence of site-specific, obesity-related metabolites, reflecting the populations’ habitual diet and lifestyle habits [37]. This was observed in a study comparing South African and Ghanaian women, in which authors concluded that lifestyle may be a key moderator of obesity [37]. These habits are modifiable risk factors, and individuals who change their metabolic profile in response to caloric restriction have significantly better retention of weight loss compared to obese individuals who have not changed it [34]. In parallel to this finding, exercise training also has the potential to alter specific intramuscular lipid intermediates, indicating that an increase in lipid utilization may prevent skeletal muscle lipotoxicity [35].

Two studies included in this review assessed nonalcoholic fatty liver disease [39,40], and two others studied cancer patients [41,42] (Table 4).

The study by Chashmniam et al. [39] evaluated the serum metabolomic profile of patients with nonalcoholic fatty liver disease (NAFLD) and healthy controls and observed alteration in 19 metabolites in NAFLD patients, particularly a reduction in precursors of some amino acids. The authors suggested that supplementation of these amino acids might be useful in the treatment of NAFLD. Supplementation with curcumin in adults with NAFLD for 8 weeks produced a decrease in inflammatory mediators and an effect in some amino acids, tricarboxylic acid cycle metabolites, gut-microbiota-derived metabolites, and bile acids [40], stressing the use of polyphenols as anti-inflammatory compounds.

Both studies with cancer patients compared patients with healthy controls. Amiri-Dashatan et al. [41] studied females with invasive breast cancer and noticed 20 significantly altered metabolites in patients, mostly amino acids and lipids. These results indicate a dysregulation in metabolic pathways, and this information can be useful for identifying diagnostic and prognostic biomarkers. Chinese researchers investigated the metabolomic profile of colorectal cancer patients and observed a clear differentiation of biomarker panel with lipid changes as the disease progressed [42].

Lifestyle and environmental factors have strong potential to positively impact the metabolomic profile of individuals, thus preventing NCDs [14]. There is evidence that long-term physical activity, but also short-term practice, plays an important role in influencing the expression of metabolites involved in NCDs development [19,20]. Furthermore, adhering to the recommended WHO diet [22], reducing red meat intake [26], and developing a welcoming palate [29] are desirable habits to prevent NCDs. Most of these healthy habits are built during childhood and adolescence, and, in this scenario, families, communities, and schools are important actors [43].

In this context, planned cities offer more opportunities for healthy practices and health promotion since green spaces make up the urban fabric. The scientific literature has exhaustively shown the benefits that urban forests bring to the physical and mental health of the population [44,45,46]. There are many authors evidencing that well-planned green spaces can prevent diseases, especially NCDs [47,48,49], with the caveat that, in low-income countries, studies involving urban green areas and health are still limited and need to be redesigned, especially to include longitudinal studies that use more robust methods [50]. In this sense, the assessment of health indicators through the lens of metabolomics can be an alternative to fill this existing gap.

This study has some limitations. The studies included reflect the publications indexed under the respective descriptors and may not represent the full picture of the publications in this area. Moreover, among the studies included in the present review, there was none addressing chronic respiratory diseases. Air pollution is under-examined and inadequately addressed by existing approaches to NCD prevention [9]. In addition to affecting the ecological systems, pollutants also affect human health, contributing to the burden of NCDs and their economic and social associated costs [51].

Efforts to equitably allocate NCD resources must include a balance of both prevention and treatment of existing cases [5]. Understanding the cost of delivering care in regions where there will be a critical increase in NCD prevalence is crucial to achieving universal health coverage and improving overall population health [5].

The use of a noninvasive or minimally invasive mode of sample collection for metabolomics analysis is a positive aspect that provides acceptability to potential participants. While studies conducted so far have provided insights on the characterization of biological pathways associated with various NCDs, other factors remain unknown and deserve future investigations to enable novel clinical, industrial, and political applications to benefit populations worldwide.

## 4. Conclusions

This systematic review analyzed several studies ranging from those on breastfeeding, muscle biopsies, and the health of night workers to numerous cases of metabolomic profiling in people with diabetes, validating metabolomics markers as predictors of inflammatory processes and NCDs.

There is still a knowledge gap about the metabolomic profile associated with NCDs in low-income countries. However, there is a need to expand the use of the metabolomics tool to other fields of knowledge, promoting new studies that associate the metabolomic profile with aspects of urban planning, since unhealthy or poorly planned environments can trigger NCDs. These novel findings may help resolve open questions about the use of urban green spaces and the benefits to physical and mental health in addition to contributing to the prevention and treatment of future cases of NCDs.

## Figures and Tables

**Figure 1 ijerph-20-06433-f001:**
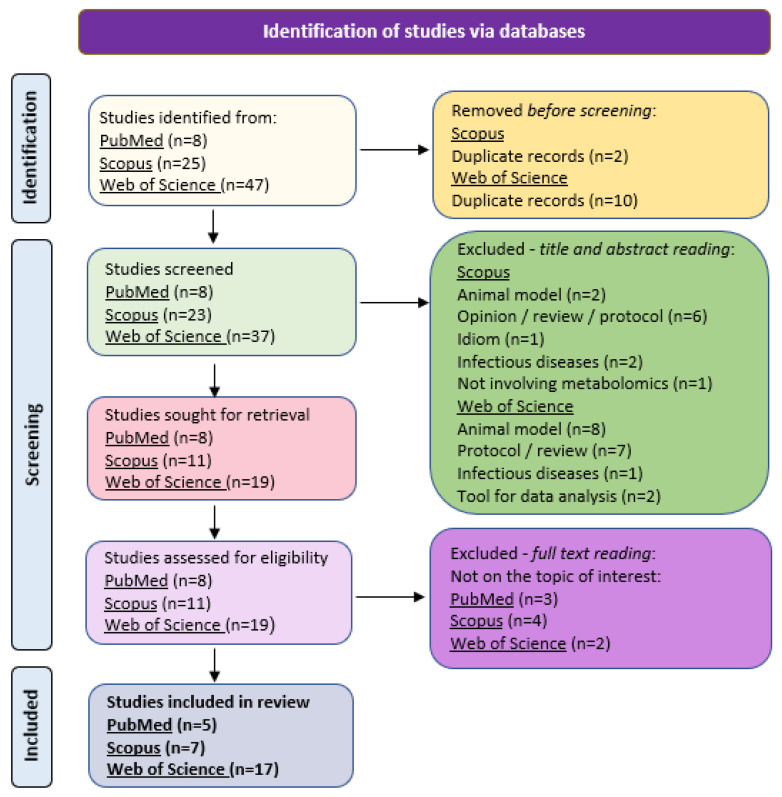
Study flowchart.

**Figure 2 ijerph-20-06433-f002:**
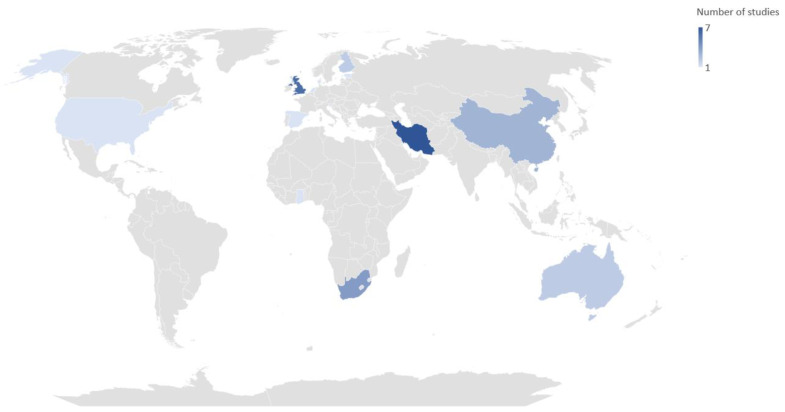
Countries of the publications included in this systematic review.

**Table 1 ijerph-20-06433-t001:** Studies involving prevention of non-communicable diseases (n = 11).

Reference	Population	Intervention	Comparison	Observation
Rauschert et al., (2017) [14]	1024 offspring followed for 20 years (Western Australian Pregnancy Cohort Study)	Retrospective study of breastfeeding duration, anthropometric measures, and metabolomic profile of plasma samples from prenatal to 20 years of age	Association between early life breastfeeding, BMI, WC, and metabolomic profile at 20 years	Metabolomic profile might be influenced by current lifestyle and environmental factors (sex, rather than breastfeeding). Physical activity, genetics, and eating habits may explain the association of maternal and paternal BMI with higher levels of waist circumference and BMI at 20 years of age
Ellul et al., (2019) [15]	1180 Australian children aged 11–12 years and 1325 parents	Cross-sectional study involving analysis of 74 blood biomarkers (amino acid species, lipoprotein subclass measures, lipids, fatty acids, measures related to fatty acid saturation, and composite markers of inflammation and energy homeostasis)	Children and adults, males and females	Aduts presented higher metabolite concentrations than children, and sex differences were also larger in adults when compared to children. There was also considerable evidence of association between parent and child measurements. Identification of risk biomarkers for several non-communicable diseases early in life can provide a target for interventions and follow up with efficacy
Drenos et al., (2016) [16]	13,285 participants of 2 UK population cohorts	Follow-up study with metabolic measures of children collected at ages 7, 15, and 17 years and mothers (at median age of 48 years), as well as women aged 60–79 years old from another cohort	APOC3 and LPL associations	The effects of *APOC3*(rs138326449) loss of function mutation in lipoprotein metabolism and its potential to affect triglyceride levels were characterized. The association with triglyceride, VLDL, and HDL levels was confirmed, and additional associations with VLDL and HDL composition, other cholesterol measures, and fatty acids were identified. Such an approach is useful for assessing drug targets, particularly in dyslipidemic individuals
Hübel et al., (2021) [17]	3107 adolescents, a subsample of a longitudinal population-based study conducted in the UK	Follow up of parent-reported eating behavior trajectory during childhood	Plasma metabolomic profile at age 16 of subgroups of eating behavior during childhood	Adolescents who used to overeat in childhood presented a metabolic profile characterized by higher lipid concentrations and chronic inflammation. This profile is a risk factor for a variety of NCDs. Children reported to undereat in childhood presented higher concentrations of glutamine in adolescence (an alternative energy substrate when diet lacks energy). Children persistently fussy about eating presented lower concentrations of BCAA (obtained from diet only)
Yu et al., (2023) [18]	163 children aged 6–14 years from Chinese boarding schools	Cross-sectional study on metabolic markers of intestinal microbiota in fecal samples	Normal weight and overweight/obesity	*Bifidobacterium*, *Alistipes*, and *Megamonas* were presented as possible biomarkers to distinguish between children who are overweight/obese and those with normal body weight. Fecal metabolites (especially fatty acids) were significantly altered in obese children. Intestinal microbiota and its related metabolites may play an important role in the development of obesity
Orrú et al., (2022) [19]	9 lifelong male football players and 9 male age-matched (64–71 years old) untrained subjects (Denmark)	Cross-sectional study with muscle biopsies	Protein and metabolite profile of veteran football players and sedentary subjects	Mitochondrial biogenesis is effectively triggered lifelong in football players. Increased concentration of molecules with anti-inflammatory properties was found in veterans when compared to controls. Structured physical activity significantly influences the expression of proteins and metabolites related to successful aging and help reduce the onset of NCDs in the elderly
Deutsch et al., (2022) [20]	20 healthy young male (10 trained and 10 untrained individuals) from Slovenia	10 days of aerobic training (60 min of supervised cycling)	Exercise performance in and urine metabolomic profile of trained and untrained individuals	Significant multivariate differences were observed in physiological characteristics between trained and untrained groups, pre and post exercise protocol. Cholate, tartrate, cadaverine, lysine, and N6-acetyllisine were the most important metabolites allowing distinguishment between trained and untrained groups. A relatively little effort in terms of exercise invested by the untrained individuals was enough to modify their urine metabolome into an indistinguishable pattern from the trained group, providing a good basis for future recommendations for health maintenance and prevention of NCDs
Harding et al., (2022) [21]	49 males, rotating factory shift workers (Spain)	Cross-sectional study involving metabolomic analysis of plasma samples	Night shift and day shift workers	Working the night shift produced association with changes in metabolites and perturbation of metabolic and biochemical pathways connected to a variety of health outcomes. Levels of several amino acids, glycerophospholipids, and one biogenic amine were higher during the night shift compared to day shift after at least a week of adaptation. Findings support the associations between night shift work and many common NCDs
Garcia-Perez et al., (2017) [22]	19 healthy volunteers, aged 21–65 years old (UK)	Dietary intervention in four 3-day inpatient periods	Urinary metabolic phenotype of subjects after 4 types of diet, varying according to compliance with WHO guidelines	Urinary biomarkers concentrations from healthy foods were significantly higher after the consumption of a diet with better compliance with WHO guidelines to prevent NCDs, reflecting an increased intake of fruits, vegetables, salmon, and chicken. Metabolic phenotyping can provide objective measures of adherence to dietary recommendations without the need for dietary surveys
Pietzner et al., (2021) [23]	11,966 men and women from the EPIC-Norfolk prospective cohort (UK)	Retrospective study of metabolomic profile of patients who died or were hospitalized by NCDs	Plasma metabolomic profile (since 1993–1997)	420 metabolites were common to at least 2 NCDs, representing 65.5% of all 640 significant metabolite–disease associations. Low-grade inflammation, obesity, smoking, impaired glucose homeostasis, lipoprotein metabolism, and liver and kidney function were identified as common actionable antecedents of NCD multimorbidity, and all of them are potentially preventable early.
Würtz et al., (2016) [24]	9778 young adults from 3 population-based cohorts in Finland	6-year follow-up study on plasma metabolomics	Individuals with and without usual alcohol consumption	Increased alcohol intake was related to cardiometabolic risk markers in multiple metabolic pathways. The metabolic signature of alcohol consumption involved molecular perturbations linked with higher and lower cardiovascular risk. Many metabolic measures displayed an optimum level at low or moderate alcohol intake.

BMI: body mass index, WC: waist circumference, NCD: non-communicable diseases, APOC3: apolipoprotein C-III, LPL: lipoprotein lipase, VLDL: very-low-density lipoprotein, HDL: high-density lipoprotein, NCDs: non-communicable diseases, BCAA: branched-chain amino acid, WHO: World Health Organization.

**Table 2 ijerph-20-06433-t002:** Studies involving diabetes, systemic arterial hypertension, and metabolic syndrome (n = 8).

Reference	Population	Intervention	Comparison	Observation
Li et al., (2022) [26]	11,432 males and females aged 40–79 years, plus a randomized crossover intervention with 12 volunteers (UK)	3-day diet intervention (red meat or nonmeat) followed by 10-day washout period (normal diet), followed by 3-day diet intervention (red meat or nonmeat)	Metabolomic analysis of plasma samples and meat consumption assessed with a 7-day dietary diary	Eleven top-ranked metabolites were associated with red meat intake, suggesting a link between this eating habit and change in these metabolites. Red meat metabolite score was associated with T2D incidence and also potentially associated with other cardiometabolic diseases
Zeng et al., (2019) [27]	476 black women with normal glucose tolerance at baseline and 144 subjects at follow up (South Africa)	Prospective cohort study, follow up for 13 years	Development of T2D, impaired glucose tolerance, or none	Women that developed T2D presented a higher baseline LPC(C18:2):LPE(C18:2) ratio, a different bile acid metabolite profile, and higher levels of leucine, along with its catabolic intermediates (i.e., ketoleucine and C5-carnitine), compared to women that remained normally glucose tolerant during the same period. These metabolite patterns can be useful for identifying and monitoring T2D risk > 10 years prior to the disease onset
Hosseinkhani et al., (2022) [28]	206 diabetic individuals and 206 healthy controls (Iran)	Cross-sectional study involving metabolomic analysis of plasma samples	Diabetic individuals and healthy subjects	Some amino acids and acylcarnitines were considered potential risk markers for diabetes. They reflect disturbances in various metabolic pathways among the diabetic population and can be targeted to prevent, diagnose, and treat this disease
Sarin et al., (2019) [29]	Subsample of adults and elderly individuals from the Finnish Dietary, Lifestyle, and Genetic determinants of Obesity and Metabolic Syndrome cohort (n = 2982) and the Estonian Biobank cohort (n = 1109)	Cross-sectional study involving assessment of food neophobia, dietary quality, and plasma metabolomic profile	Association of food neophobia with dietary quality and health-related biomarkers	Food neophobia was significantly associated with many health-related biomarkers, most strongly with *ω*-3 fatty acids. It was also associated with poor overall dietary quality in Finnish individuals. A high level of food neophobia was associated with increased incidence of T2D in the Finnish cohort, whereas an increased incidence of CHD was detected in the Estonian Biobank cohort
Arjmand et al., (2022) [30]	403 non-diabetic adults (aged 18–75 years) from a population-based study (Iran)	Cross-sectional study involving plasma metabolite profiling	Patients with and without IR	A specific metabolomic profile perturbation is associated with IR. There was a strong positive association between serum BCAAs (valine and leucine), AAAs (tyrosine, tryptophan, and phenylalanine), alanine, and C0 (free carnitine) and IR, while C18:1 (oleoyl L-carnitine) was negatively correlated with IR
Strauss-Kruger et al., (2022) [31]	910 healthy young adults and 210 older adults from 2 independent cohorts in South Africa	Cross-sectional study involving anthropometric measures, dietary intake, and blood pressure and urine metabolomic profiling	Normotensive persons and individuals with masked hypertension	Significant differences were observed between the metabolomic profiles of normotensive and hypertensive adults. They may reflect different stages in the alteration of BCAA metabolism. These changes may be modulated by central adiposity, which indicates that metabolic dysfunction may be an underlying contributor to the etiology of masked hypertension
Arjmand et al., (2023) [32]	1200 Iranian persons aged over 18 years old	Cross-sectional study involving measure of plasma concentrations of 30 acylcarnitines and 20 amino acids using a targeted approach with flow injection tandem mass spectrometry	Persons with normal blood pressure, elevated blood pressure, stage 1 hypertension, and stage 2 hypertension	After adjustment for confounders, 5 metabolites were considered good risk markers for stage 2 hypertension. High plasma levels of carnitines, various acylcarnitines, and low plasma levels of glycine may play a crucial role in the development and progression of hypertension
Taghizadeh et al., (2023) [33]	1192 participants from a large-scale study conducted in Iran, 529 with MetS and 663 without MetS	Cross-sectional study on circulating levels of metabolites (amino acids and acylcarnitines)	Patients with and without MetS	Changes in amino acid and acylcarnitines profiles were observed in patients with MetS. The alteration in circulating levels of acylcarnitines and amino acids followed the increase in MetS component number. Amino acid and acylcarnitines profiles can provide valuable information on evaluating and monitoring MetS risk

T2D: type 2 diabetes, LPC: lysophosphatidylcholine, LPE: lysophosphatidylethanolamine, NCD: non-communicable diseases, IR: insulin resistance, BCAA: branched-chain amino acid, AAA: aromatic amino acid, MetS: metabolic syndrome.

**Table 3 ijerph-20-06433-t003:** Studies involving obesity (n = 6).

Reference	Population	Intervention	Comparison	Observation
Tareen et al., (2020) [34]	57 Caucasian participants of a weight loss study conducted in the Netherlands aged 32–67 years	Low- or very-low-calorie diets for 12 and 5 weeks, respectively, plus follow up	Weight loss period and maintenance period	Obese individuals who changed their metabolic profiles in response to caloric restriction had a significant retention of lost weight compared to individuals who did not change it. Cellular metabolism was downregulated during weight loss, with gene expression of all major cellular metabolic processes being lowered during weight loss and maintenance. In parallel, gene expression of immune-system-related processes involving interferons and interleukins increased
Mendham et al., (2021) [35]	35 sedentary obese black South African women aged 20–35 years old	Secondary analysis of a randomized, controlled trial with 12 weeks of combined aerobic and resistance training or control (metabolomic, lipidomic of muscle samples)	Exercise group and sedentary controls	Exercise training altered specific intramuscular lipid intermediates which were associated with content-driven increases in mitochondrial respiration and not whole-body insulin sensitivity or GLUT-4 protein content. Exercise increases lipid utilization in the more bioenergetically active organelles and membranes, which may prevent future skeletal muscle lipotoxicity
Yan et al., (2023) [36]	86 healthy Chinese young adults, 31 men and 55 women	Cross-sectional study involving anthropometry measures, food and drink frequency and lifestyle habits questionnaires, metabolomic and lipid profile analysis of plasma samples	Association between intake of sugar-sweetened beverages and obesity-related, gut-microbiota-related metabolic markers and blood lipids	SSBs were positively associated with obesity-related markers and blood lipids. In contrast, presweetened coffee was negatively associated with the obesity-related traits. A total of 79 metabolites were associated with both SSBs and metabolic markers, particularly obesity markers. Branched-chain amino acid catabolism and aminoacyl-tRNA biosynthesis were identified as links between SSB intake and metabolic health outcomes, relevant to non-communicable diseases
Dugas et al., (2016) [37]	Black women from USA (n = 69), South Africa (n = 97), and Ghana (n = 82)	Cross-sectional study involving BMI, physical activity, dietary intake, and metabolomic analysis of plasma samples	Normal weight and obese black women	A common amino acid metabolite profile was associated with obesity in black women from the USA and South Africa but was not detected in Ghanian women. Site-specific, obesity-related metabolites associated with obesity included intermediates in lipid and carbohydrate metabolism, reflecting the populations’ habitual diet. Local environment (and consequently lifestyle factors) seems to be a key moderator of obesity
Bagheri et al., (2019) [38]	200 Iranian obese adults and 100 healthy controls	Cross-sectional study involving plasma metabolomic profiling, arterial blood pressure, anthropometric data, dietary intake, and physical activity	Obese and healthy controls	A metabolomic profile containing 9 amino acids and 10 polar lipids may serve as a potential biomarker of adult obesity. Obese patients had significantly higher blood pressure than controls. No significant associations between metabolites and energy intake or physical activity were observed
Bagheri et al., (2018) [2]	285 Iranian adults aged 18–50 years old	Cross-sectional study with anthropometric data, blood pressure and biochemical measurements, dietary intake, physical activity, and plasma metabolomic analysis	MHO, MUHO, and normal-weight metabolically healthy subjects	Some metabolites discriminated MHO from MUHO and highlighted MHO- and MUHO-related metabolite patterns associated with cardiometabolic biomarkers. Two MHO-associated and three MUHO-linked factors related to more than 75% of the assessed cardiometabolic parameters. BCAAs, glutamic acid, tyrosine, and a specific pattern of lysophosphatidylcholines were associated with risk of both MHO and MUHO phenotypes. A particular pattern of amino acids and choline-containing phospholipids may aid in the identification of metabolic health among obese patients

SSBs: sugar-sweetened beverages, BMI: body mass index, MHO: metabolically healthy obesity, MUHO: metabolically unhealthy obesity, BCAA: branched-chain amino acids.

**Table 4 ijerph-20-06433-t004:** Studies involving cancer and nonalcoholic fatty liver disease (n = 4).

Reference	Population	Intervention	Comparison	Observation
Chashmniam et al., (2019) [39]	36 healthy controls and 37 patients with NAFLD (Iran)	Cross-sectional study involving clinical data, anthropometric measures, and serum metabolomic profiling	Metabolomic profile of healthy controls and patients with NAFLD	Nineteen metabolites were altered in NAFLD patients when compared to control group. Metabolomics biomarkers revealed changes in some amino acids and their derivatives, bile acids, short-chain fatty acids, and tricarboxylic acid cycle intermediates in subjects with NAFLD compared to healthy controls. Precursors of glutathione, glutamine, serine, and glycine were reduced in NAFLD patients. Supplementation of these amino acids might be useful for treatment of this condition. Decreased levels of short-chain fatty acids could be result of altered levels of gut microbiota. Supplementation with probiotics is also recommended
Chashmniam et al., (2019) [40]	45 adults, NAFLD outpatients from an Iranian hospital	8 weeks of curcumin supplementation, followed by plasma metabolomic profiling, anthropometric, and clinical measures	Curcumin supplementation (50 mg/day) and control (placebo) group	Curcumin intake for 8 weeks in patients with NAFLD had effects in some amino acids, TCA cycle metabolites, gut-microbiota-derived metabolites, and BAs. TCA cycle was the metabolic pathway found to be changed most significantly between groups. A decrease in oxidative and inflammatory mediators was observed with the use of this polyphenol
Amiri-Dashatan et al., (2022) [41]	22 females with invasive ductal breast cancer aged 30–65 years old (Iran) and 10 healthy controls	Cross-sectional study involving plasma metabolomic profiling	Women with invasive ductal carcinoma and healthy controls	The 20 significantly altered metabolites in cancer patients’ samples mostly belonged to amino acids and lipids. Results indicated significant dysregulation of metabolic pathways in breast cancer patients. Their metabolite profile was related to the reprogramming of amino acids and lipid metabolism, mainly arginine and proline metabolism. Metabolomics profiling in breast cancer patients might be useful for identifying biomarkers related to diagnostic and prognosis and also for monitoring treatment
Li et al., (2013) [42]	52 CRC patients and 52 healthy controls (China)	Cross-sectional study involving metabolomic profile of serum samples	CRC patients in early stage, late stage, and healthy subjects	Analyses revealed a successful differentiation of a biomarker panel between CRC patients and healthy controls and revealed signature lipid changes as CRC progresses. The determination of changes during disease progression may provide therapeutic opportunities, such as nutritional support and disease condition monitoring, as well as evaluation of the effectiveness of various treatment options

NAFLD: nonalcoholic fatty liver disease, TCA: tricarboxylic acid, BA: bile acids, CRC: colorectal cancer.

## Data Availability

Not applicable.

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
