# Peer review of "Environmental Planning and Non-Communicable Diseases: A Systematic Review on the Role of the Metabolomic Profile"

_ijerph, 2023, doi:10.3390/ijerph20146433_

Round 1
Reviewer 1 Report
Thank You for the opportunity to review the manuscript titled: “Environmental Planning and Non-Communicable Diseases: a Systematic Review on The Role of Metabolomic Profile” for the IJERPH journal.
This manuscript was a solid, informative and enjoyable read. The authors present us with a review of the potential uses of metabolomics in NCDs with references to urban planning. This is a rather novel topic and one that is bound to become an important factor in environmental planning (especially urban planning) in most societies as we struggle to keep up with the burdens of NCDs.
There is really little fault that can be found in this manuscript, and I present those points in a few of the following lines. These revisions are merely language- or typing-related, so therefore I wholehartedly recommend this article to be accepted for publication in IJERPH after a few minor revisions.
Line 104: The sentence “The first search resulted in zero publications after exclusion criteria.” calls for at least some explanation. What did the authors do differently in the secound round of search to yield more results?
Lines 121-2: Two following sentences “Eleven studies involved NCDs prevention (Table 1). Among them, five studied children or adolescents [14-18].” should be merged into one as the second sentence now lacks predicate.
In Table 1, row “Rauschert et al. (2017)”, column “population” should be “for 20 years” instead “by 20 years”.
Line 155: There is a typo in “byurine”. Please, correct.
Line 158: There is a typo in “adiet”. Please, correct.
Sincerely,
The reviewer
The English language use is solid and apart from a few typos and several mistakes (which I pointed out in the review itself), I found no other issues with it.
Author Response
Thank you for the compliments on the manuscript.
Line 104: The sentence “The first search resulted in zero publications after exclusion criteria” was clarified as is highlighted in the text. We meant that the search “metabolomics” AND “noncommunicable diseases” AND “air pollution resulted in zero publications, and the next search, removing the keyword "air pollution" (described in the methods section), presented better results.
Lines 121-2: The sentences “Eleven studies involved NCDs prevention (Table 1). Among them, five studied children or adolescents [14-18].” were merged into one as suggested.
In Table 1, row “Rauschert et al. (2017)”, column “population” was changed to “for 20 years” instead of “by 20 years”.
Line 155: The typing mistake “byurine” has been corrected.
Line 158: The typing mistake “adiet” has been corrected.
Reviewer 2 Report
The manuscript "Environmental Planning and Non-Communicable Diseases: a Systematic Review on The Role of Metabolomic Profile" by Natalia Cristina de Oliveira et al is interesting.
However, major comments are found on this manuscript:
- State the registration number of PROSPERO on the main manuscript.
- "PRISMA checklist" must be included as a supplementary file.
- Why only use 3 databases/platforms ? should be at least 5 platforms/databases.
- the contents of the table in the "Observation" section must be paraphrased in the author's own words; because it overlaps heavily with publications.
- "Figure 2. Countries of the publications included in this systematic review." produced by the author himself? if so, what was it produced with?
- This section has no statement from the authors:
Author Contributions:
Funding:
Institutional Review Board Statement:
Informed Consent Statement:
Data Availability Statement:
Conflicts of Interest:
Extensive editing of English language required. This manuscript must be proofread by native speakers.
Author Response
The registration number in PROSPERO was not received yet, however, the protocol receipt number is 446279.
"PRISMA checklist" was included as a supplementary file.
We used these 3 databases because they are the most important ones in the health area, embracing journals with severe editorial policies, which reinforces data credibility.
We have made corrections in the "Observation" columns.
Figure 2 was produced by the authors using microsoft excel.
Author Contributions, Funding and Conflicts of Interest were included, but no Institutional Review Board Statement or Informed Consent Statement was necessary for this study.

Reviewer 3 Report
I loved how the authors presented the introduction of NCD's and Metabolomics as playing a significant role for environmental health agencies to identify and concentrate on resources to improve overall health. No comments on methods, materials section. As far as the studies included in the analysis involves wide range from 2013 to 2017 ... so on major flaw i find is these studies were more of a causual relationship and did not factor into the fact about new pharmacotherapies in the management of diabetes, hypertension, metabolic syndrome and non-alcoholic fatty liver disease.
I would like to see if the authors can modify their search criteria to come up with more recent studies to make your scientific evidence more stronger. Also to check for more RCT trials in the analysis.
Also no mention of gut microbiome which seems to be an emerging player in altering metabolomics in NCD'S.
Minor english editing is required to make the paper flow better to get interest of your audiences.
Author Response
Our search criteria did not impose publication date limit, as highlighted in lines 94-95. Studies found in tables cover dates up to 2023.
Gut microbiome was mentioned, as suggested, and is highlighted n the manuscript.
Reviewer 4 Report
This manuscript looks interesting. However, the results and discussion sections do not have a core idea, and there is a vague message. A minor amendment is therefore proposed.
1. Please consider the rationality of the keyword “urban planning”.
2. “Environmental Planning” and “Non-Communicable Diseases”: They do not seem to be juxtaposed in the manuscript, and the environmental planning is an assumption, and the related narrative is weak. It is suggested that the title be modified.
3. Figure 2 should be annotated or labeled with more information so that the reader can understand it intuitively.
4. Lines 35-39: “Insights into 35 biological pathways associated with NCDS” are not clearly stated in the manuscript.
5. Low-income countries are not specified in the manuscript.
6. Results and discussion: This section lacks a core main idea.
7. Lines 204-205: The content here is vague, which is not conducive to the reader to obtain effective information.
8. Lines 281-283: The expression of the road map is abrupt and is not mentioned in the manuscript.
The author needs to carefully check grammar and spelling errors again.
Author Response
- The keyword “urban planning” has been changed to "environmental planning".
- The suggestion to change the title was not incorporated in the text, as other reviewers complimented the subject we are discussing in the present manuscript.
- More info on Figure 2 were highlighted in lines 117-121, instead of doing it in the figure caption, as it would be extensive.
- Lines 35-39: the sentence was re-written, excluding the number "35" from the middle of it.
- Low-income countries were the ones listed on reference 5, as highlighted in lines 55-57.
- Results and discussion: This section has been revised.
- Lines 204-205: The sentences were re-written.
- Lines 281-283: The expression "road map" was changed.